# Development of ZSM-22/Polyethersulfone Membrane for Effective Salt Rejection

**DOI:** 10.3390/polym12071446

**Published:** 2020-06-28

**Authors:** Nyiko M. Chauke, Richard M. Moutloali, James Ramontja

**Affiliations:** 1Department of Chemical Sciences, Faculty of Science, University of Johannesburg, Doornfontein 2028, Johannesburg, South Africa; nyikomau@gmail.com; 2DSI/MINTEK Nanotechnology Innovation Centre-Water Research Node, University of Johannesburg, Doornfontein 2028, Johannesburg, South Africa

**Keywords:** zeolite ZSM-22, polymer membranes, polyethersulfone, salt rejection

## Abstract

ZSM-22/polyethersulfone membranes were prepared for salt rejection using modelled brackish water. The membranes were fabricated via direct ZSM-22 incorporation into a polymer matrix, thereby inducing the water permeability, hydrophilicity and fouling resistance of the pristine polyethersulfone (PES) membrane. A ZSM-22 zeolite material with a 60 Si/Al ratio, high crystallinity and needle-like morphologies was produced and effectively used as a nanoadditive in the development of ZSM-22/PES membranes with nominal loadings of 0–0.75 wt.%. The characterisation and membrane performance evaluation of the resulting materials with XRD, BET, FTIR, TEM, SEM and contact angle as well as dead-end cell, respectively, showed improved water permeability in comparison with the pristine PES membrane. These ZSM-22/PES membranes were found to be more effective and superior in the processing of modelled brackish water. The salt rejection of the prepared membranes for NaCl and MgCl_2_ was effective, while they exhibited quite improved water flux and flux recovery ratios in the membrane permeability and anti-fouling test. This indicates that different amounts of ZSM-22 nanoadditives produce widely divergent influences on the performance of the pristine PES membrane. As such, over 55% of salt rejection is observed, which means that the obtained membranes are effective in salt removal from water.

## 1. Introduction

The global demand for freshwater is increasing every day due to high population growth and industrial development. Brackish water and seawater are abundant but have high salt concentrations and therefore need to be purified [1,2,3]. To effectively utilise these available water sources, sustainable and efficient water purification technologies such as membrane filtration and ion exchange resins are needed. Based on this consideration, there is a need to design and develop new functional materials for the desalination of saltwater using membranes. Nanocomposite membranes with low surface fouling, superior physicochemical integrity, enhanced water flux and high solute rejection have been identified as potentially promising separation materials [4,5,6]. Furthermore, membranes offer high efficiency in the separation and filtration processes [6,7]. However, due to the polymer membrane’s shortcomings, such as fouling, high hydrophobicity and short functional lifetime, more work needs to be done in the redesigning and modifying of nanocomposite membrane materials [7,8,9,10,11]. In consideration of these limitations, the scope for membrane modification has widened for different applications, such as the desalination and distillation of radiative waste and salt solutions [12,13,14].

Polyethersulfone (PES) membranes have been universally used and accepted as appropriate polymers for use in separation membranes, owing to their high chemical, mechanical and thermal stability under highly pressurised separation systems [15,16]. They have gained widespread use in microfiltration (MF) and ultrafiltration (UF) membranes, but showed poor performance in extremely high pressurised systems such as nanofiltration (NF) and reverse osmosis (RO) [17,18,19]. Their relatively poor performance in NF and RO systems is attributed by some in the literature to their weak mechanical stability as a consequence of their high amorphous and lower hydrophilic character [20,21,22,23]. As a result, PES membrane utilisation in NF/RO application is highly prone to fouling, which is considered the main limitation to the use of PES membranes in highly pressurised separation processes [23,24,25,26].

In consequence, the challenges associated with membrane quality also appear to be influenced by the cost of membrane preparation, maintenance and lifetime. As membrane surface hydrophobicity promotes fouling, this later leads to the low flux, permeability and lifetime of the membrane [10,27,28,29]. However, typical surface modification of PES by the inclusion of highly hydrophilic nanoadditives such as hollow halloysite nanotubes (HNTs), graphene oxide (GO), titanium dioxide (TiO_2_), silicate, etc., in the polymer matrix is a promising solution [29,30,31,32,33]. However, the challenge remains regarding how to identify effective nanoadditives to mitigate membrane fouling. Suitable nanoadditives (in terms of particle size and hydrophilicity) are still needed for the development of highly functional membrane materials [8,25,33]. In this consideration, synthesising new polymer materials is not a priority given the opportunity to modify already existing polymer materials.

Therefore, the current focus on membranes research should be directed to designing and developing effective nanocomposite membranes with low surface fouling, superior physicochemical integrity, enhanced water flux and high solute rejection [34]. As such, research on polymeric membrane material and membrane technology maintained its focus on membrane modification to produce materials with intrinsic composite features [2,35,36,37]. This involves a straightforward process of conjoining or blending polymeric material with hydrophilic nanoadditives such as nanoparticles, carbon nanotubes and zeolites [1,23,38,39]. Typically, zeolite materials have demonstrated the ability to enhance membranes’ water flux, permeability and antifouling characteristics, as reported in several studies [40,41].

Zeolites are alumina silicates materials widely used in ion exchanging, hydrocracking, adsorption and catalysis [42,43]. Recently, they have gained much consideration in the derivation of composite membrane materials owing to their external surface properties such as silinol, their negatively charged framework, shape selectivity and pore size exclusion [44,45,46,47]. These features promote high interaction between the polymer and additive as well with the solute cations and anions. Also, their hydrophilic character has earned them their definitive usage as inorganic nanoadditives for composite membrane development [48,49,50,51,52]. This study has therefore identified a need to develop effective materials due to the limited work associated with PES membranes for typical NF or RO applications. This report demonstrates that, with the judicious choice of fillers, imparting additional functionalities to UF membranes does result in nanocomposite polymer membranes with relatively high salt rejection for potential use in brackish water treatment. Herein, ZSM-22 is embedded in the PES matrix to result in NF membranes with up to 55% and 65% rejection rates for NaCl and MgCl_2_, respectively.

## 2. Materials and Methods

The following chemicals and reagents were purchased from Sigma-Aldrich (Johannesburg, South Africa) and used without any purification: tetraethyl orthosilicate (TEOS), aluminium sulphate octadecahydrate ((Al_2_(SO_4_)_3_·18H_2_O), hexamethylenediamine (HMDA), potassium hydroxide (KOH), sodium hydroxide (NaOH), hydrochloric acid (HCl), nitric acid (HNO_3_), sodium chloride (NaCl), magnesium chloride (MgCl_2_), bovine serum albumin (BSA), nitrogen gas (N_2_), polyethersulfone (PES) 3 mm nominal granule size and N-methyl-2-pyrrolidone (NMP).

ZSM-22 zeolite materials were synthesised using procedures previously reported by Marler [53] and Ernest [54], with some modification as follows: A clear aqueous solution was prepared by dissolving KOH (1.252 g) and Al_2_(SO_4_)_3_·18H_2_O (0.743 g) in deionised water (49 mL) under magnetic stirring for 1 h to produce solution A. In a separate beaker, HMDA (10.97 mL), as a structure-directing agent (SDA), was dissolved in deionised water (20 mL) and allowed to stir for about 1 h to produce solution B. Thereafter, solution B was transferred into solution A under stirring, resulting in solution C. Hydrolysed TEOS (29 mL), obtained from the addition of deionised water (69 mL) after stirring for 1 h, was added dropwise (at a rate of 3.27 mL/min) into solution C [55,56]. After 1 h, a solution containing a final gel-like product with molar composition of 60SiO_2_:Al_2_O_3_:9KOH:27DAH:3600H_2_O was obtained. Thereafter, the gel was transferred into a 150 mL stainless steel autoclave and heated to 160 °C for 144 h for crystallisation to occur. The resulting white crystals were recovered by filtration and subsequently washing with distilled water until the pH of the filtrate was seven. The obtained solid samples were air-dried overnight and calcined at 550 °C for 24 h to remove the SDA.

The ZSM-22/PES membranes were fabricated via direct incorporation of ZSM-22 nanoadditives prepared using TEOS into the polymer solution using a phase inversion method. PES granules (18 g) were dried at 80 °C for 24 h before dissolution in NMP (82 mL) solvent. This was followed by the addition of the required amount (0–0.75 wt.%) of the ZSM-22 powder into a casting solution and allowed to stir for 24 h until a homogenous mixture was attained. The casting solution was allowed to degas under a vacuum for 24 h to dissipate the trapped air in the solution. For effective nanoadditive distribution in the solution mixture, the casting solution was subjected to an ultrasonication for about 3 h before casting. Then, the solution was cast using a casting knife set at a 200 μm air gap, then allowed to stand for 30 s in the air before immersion into the coagulation (deionised water) bath for 15 min. Thereafter, the membranes were transferred into a clean deionised water bath and allowed to cure for 24 h, then stored in deionised water kept in the refrigerator for further analysis and assessment.

The X-ray diffraction patterns of different samples were recorded using a PANalytical PW 3050/60 diffractometer (XPert-Pro, Almelo, The Netherlands) with PSD Vantec-1 detectors and Cu Kα radiation (λ = 1.5406 Å), at a scan step size of 0.025°. The time/step was in seconds, at a scan speed of degree/second. The BET surface measurements of zeolite ZSM-22 materials were carried out on an automated gas adsorption and surface area analyser, Micrometrics TriStar II Plus Version 3.00 (Micromeritics, Norcross, GA, USA) and Porosity Analyser 3000 (Micromeritics, Norcross, GA, USA) equipped with Win 3000 software at −195.8 °C. The surface area and the pore size and volume of the material were determined by single point analysis. The attenuated total reflectance Fourier-transform infrared spectroscopy (ATR-FTIR) spectra of the resulting samples in this study were attained using a Perkin Elmer Spectrum 100 FTIR spectrometer with the scan range of 400–4000 cm^−1^ at a resolution of 4 cm^−1^ and over an average of 16 scans. The analysis was performed using Bruker Vector 22 mid-IR spectroscopy (Bruker, Karlsruhe, Germany). Transmission electron microscopy (TEM) analysis of ZSM-22 zeolite materials was done on a Jeol JEM 2100 transmission electron microscope (Tokyo, Japan) under a bright-field at 120 kV. A precise amount of the sample was sonicated in about 5 cm^3^ of ethanol under 60 W ultrasonic bath for about 10 min. Then a tiny drop of the suspension was placed on a coated copper grid. The grid was dried in the air before mounting on the TEM sample holder for analyses. The morphological structure of the samples was scanned at 5 kV using lowest beam current of the scanning electron microscopy (SEM), which achieved the optimum resolution at a specific magnification. A small amount of zeolite powder from a piece of the membrane was mounted on a sample holder using a carbon tap and was carbon-coated before analysis. In evaluating the membrane surface hydrophilicity or hydrophobicity, a contact angle goniometer (G10, KRUSS, Hamburg, Germany) was used. The water contact angle (°) of the prepared membranes was measured at a constant room temperature and 50% humidity using the sessile drop method.

Stock solutions of NaCl and MgCl_2_ (ca. 1.00 g each) were dissolved in a 1000-mL volumetric flask using deionised water to obtain 1000 ppm solutions. The standard solution was prepared by dilution of the stock solution: typically, a 250-ppm standard solution of NaCl or MgCl_2_ was prepared by diluting 125 mL of 1000 ppm stock solution in 500 mL volumetric flask. From this standard solution, 50 ppm salt solution was prepared by diluting 20 mL of 250 ppm standard solution into a 100 mL volumetric flask and kept in the refrigerator at 4 °C to avoid solution decomposition for further analysis.

The ZSM-22/PES membrane performance indicators such as pure water flux (deionised water), solute rejection (50 ppm of NaCl or MgCl_2_) and protein fouling (using 1000 ppm BSA) were evaluated using a Sterlitech (Kent, OH, USA) dead-end stirred cell with an effective surface area of 19.63 cm^2^. The permeate flux was defined as:(1)$$P_{WF}=\frac{Q}{A\times t}$$
where *Q* is the volume of the pure water permeate (L), while *A* is the effective surface area of the membrane (m^2^), and *t* is the time (h) taken for the permeate. The permeation measurements in the study of all membranes were at room temperature. Preliminary, the water flux of each membrane was compacted until a steady state was reached under the condition of pre-compacting pressure of 1 bar for about 1 h. Following pressure was decreased to the normal operating pressure of 0.8 bar and the pure water flux (*P_WF_*) was measured. After the pure water flux test measurement, a solution of 1000 ppm BSA was transferred in the dead-end, followed by compacting and then permeation flux measurement at a similar pressure. Thereafter, the fouled membranes were then rinsed with deionised water through backwashing for 1 h, and the pure water fluxes of these backwashed membranes were retested again.

## 3. Results and Discussion

### 3.1. XRD Analysis

The XRD patterns of the zeolite ZSM-22 material and ZSM-22/PES membranes are shown in Figure 1a and Figure 1b, respectively.

As shown in Figure 1a, the XRD pattern of ZSM-22 crystal sample exhibits characteristic peaks of a Theta-1 (TON) topology of ZSM-22 material, which is in agreement with the database of zeolite structures as reported in the literature [57,58]. This material presents five peaks at 2θ values of 8.18°, 20.30°, 24.20°, 24.52° and 25.59°, respectively, corresponding to the ZSM-22 structure [59,60]. This is indicative of the successful synthesis of the highly crystalline ZSM-22 material when using TEOS as silica source at 160 °C for 144 h with an Si/Al ratio of 60. The obtained ZSM-22 zeolite material was then directly incorporated into the pristine PES casting solution, resulting in a series of composite membranes containing ZSM-22. It is observed that the XRD patterns of ZSM-22/PES membranes with to 0 wt.% loadings exhibit a mainly amorphous phase with one broad characteristic peak of a typical PES phase (Figure 1b). The incorporation of ZSM-22 nanoadditives resulted in the XRD patterns exhibiting the diffraction peaks of the ZSM-22 (Figure 1b). For instance, the pattern of the membrane containing 0.1% ZSM-22 had a peak at 2θ of 8.02° (of (110) plane) on the broad base of the amorphous PES membrane, which is indicative of the presence of ZSM-22 zeolite nanoadditives. The slight peak shift to lower 2θ values was observed, attributed to the inclusion of the polymer chains into the zeolite crystal planes. The shifts indicate a physical interaction between ZSM-22 crystallites and PES polymer chains, which is a positive observation, as it will result in minimal zeolite leaching from the membranes during application. Additional zeolitic peaks were observed at 9.73°, 12,69°, 20.30°, 24.66° and 25.74°, attributed to (021), (200), (021), (240) and (400) planes, with a slight shift to higher 2θ values as the amount of ZSM-22 was increased. The overall amorphous character of the composite membranes indicates that the addition of the zeolites did not alter the membrane structure. 

### 3.2. BET Analysis

The ZSM-22 material was analysed using BET to estimate the pore size distribution of the nanoadditives for effective salt polymer and solute chain interaction upon membrane formation and application, respectively. The N_2_ adsorption–desorption isotherms (with inserted textural properties) of ZSM-22 materials along with their pore size distribution are shown in Figure 2.

The isotherm indicates that the ZSM-22 material produced here have a significantly developed the micropore system in agreement with the reported literature results for zeolite materials [61,62,63]. This is illustrated by the amount of N_2_ uptake at a relative pressure P/P_0_ < 0.1, which represents the amount of micropore in the ZSM-22 structure as shown in Figure 2a. The material exhibits average pore size (P_S_), pore volume (P_v_) and BET surface area (S_BET_) of 2.814 nm, 1.260 cm^3^ and 228.700 m^2^/g, respectively. The existence of highly ordered micropores on the structure can be manifested by the constant N_2_ uptake at 0.1 < P/P_o_ or P/P_o_ > 1.0 relative pressure. The structure also exhibits absent hysteresis loops, suggesting a typical formation of a highly ordered and well-defined structure. Meanwhile, the BJH pore distributions between 20 Å and 100 Å (Figure 2b), respectively, suggest the coexistence of mesoporous structure hence an increase N_2_ uptake at P/P_o_ ≥ 1 (Figure 2a) can be observed. These results further indicate that the obtained ZSM-22 zeolite structure has a typical hierarchical framework of inter-crystalline mesopores, within micropore nanorods, among self-assembled ZSM-22 nanorods.

### 3.3. ATR-FTIR Analysis

An ATR-FTIR technique was used to study the functional groups of the synthesised zeolite material and prepared membranes. Figure 3 shows the ATR-FTIR spectra of ZSM-22 zeolite materials hydrothermally synthesised using TEOS as silica source at Si/Al ratio of 60 and the ZSM-22/PES membranes.

As shown in Figure 3a, with regard to the synthesised ZSM-22 zeolite material using TEOS as silica source, two main peaks, appearing at 1050 cm^−1^ and 765 cm^−1^, were related to the Si-O-Si and Si-O symmetric and asymmetric vibrations, respectively. Meanwhile, the shoulder peaks at 1200 cm^−1^ and 700 cm^−1^ could be attributed to the Al-OH and Si-O-Al bending vibration, respectively [64,65]. These further confirm that the resulting structure possesses a negatively charged framework. This is probably due to the plausible isomorphous substitution of Si^4+^ and al^3+^ during calcination in the zeolite framework. Small bands at around 1710 cm^−1^ can be traced to the insignificant vibration of Al-O-Al, amounting to lower aluminium species in the zeolite structure [65,66]. Meanwhile, Figure 3b displayed identical FTIR spectra of ZSM/PES membranes. Their spectra exhibited no bands attributable to ZSM-22 inclusion in the membranes, probably due to low amounts of ZSM-22 (typically 0.1–0.75 wt.% loadings) incorporated into the PES matrix. Another explanation for their absence might be that these bands, specifically the symmetric and asymmetric peaks of ZSM-22, are buried within the relatively more intense vibration bands from the PES polymer matrix.

### 3.4. Morphological Analysis

The morphology of ZSM-22 zeolite obtained from SEM and TEM are presented in Figure 4. As observed from the SEM micrograph (Figure 4a), the zeolite ZSM-22 exhibits agglomerates of needle-like and rod-like structures. The observed shapes are attributed to the effect of HMDA used as a structure-directing agent. Moreover, the TEM (Figure 4b) image reveals that the resulting ZSM-22 material consists of nanorods, in agreement with the SEM analysis. The observed morphologies from both SEM and TEM analyses are in line with previous reports [67]. Then, the nanocomposite membranes were also analysed using the SEM technique. 

The membrane surface and cross-section of ZSM-22/PES nanocomposite membranes are shown in Figure 5. The surface morphology (Figure 5a–e) of the membranes all exhibited a porous nature, as expected. Furthermore, as the quantity of ZSM-22 was increased, surface nodules started to emerge, indicating that the zeolite crystallites were near or on the surface layer. The cross-section also showed a typical morphology of finger-like pores with microvoids of varying shapes depending on the amount of nanoadditives used. In general, the cross-section micrographs (Figure 5a’–e’) exhibit a compact thin selective layer and porous sublayer typical of NF membranes produced through phase inversion. There is an observable transformation of both the surface and cross-section in both the surface layer and sublayer (i.e., morphology of macropores and wide micro-voids) as zeolite nanoadditives were increased. This is evident from the respective decrease and increase in pore size and pore quantity upon ZSM-22 addition. In consequence, the microvoids of the resulting membranes in the corresponding cross-section micrographs are slightly compressed/reduced in size with increasing zeolite addition, in agreement with the corresponding surface micrographs, indicative of the effect of zeolite inclusion into the polymer matrix and in agreement with reports from the literature [68,69,70].

Generally, SEM studies show that the incorporation of ZSM-22 nanoadditives into the polymer matrix has influenced the membrane formation mechanism and the final structure of the prepared membranes during the phase inversion process. As such, the incorporation of hydrophilic ZSM-22 into the PES polymer matrix resulted in reduced membrane surface pore sizes (Figure 5a–e) with a concomitant increase in pore density compared to the pristine PES membrane. This observation of surface pore size reduction and the relatively decreasing porosity of the top dense layer with increasing filler loading (Figure 5a’–e’) confirms this trend towards tighter membranes. These observations are in line with the reported expectations for hydrophilic fillers that generally result in membranes with smaller surface pores and increasing filler content [71,72]. 

### 3.5. Hydrophobicity and Hydrophilicity Analysis

A water contact angle was employed to determine the hydrophobicity/hydrophilicity character of the membranes. This technique is widely used for a quick and cheap way to estimate surface hydrophilicity or hydrophobicity even with its known limitations [71,72,73]. The limitations are, in some cases, due to the water droplet penetrating the surface into the micro-voids of the membrane gradually because of capillary force contact with the membrane. Generally, the water contact angle decreases with increasing surface hydrophilicity, indicative of the wettability of the permeable membrane [74,75]. As such, the membrane contact angle with good hydrophilicity should decrease more rapidly, in theory, when the pore size and morphology are similar for a series of membranes. 

Figure 6 shows the water contact angles of prepared membranes with varying zeolite wt.% loadings. As shown in the figure, the water contact angles of the membranes containing ZSM-22 were smaller than that of the pristine PES membrane. The water contact angle decreased as the amount of nanoadditives was increased. This decrease in contact angle with increasing nanoadditive loadings indicates that ZSM-22 imparts hydrophilicity to the membrane surface. This was expected as ZSM-22 is known to be a hydrophilic zeolite [76,77].

### 3.6. Flux and Rejection Analysis

The composite membranes were further assessed for filtration performance. In this regard, pure water flux, flux recovery ratio, as well as solute rejection, were measured for each membrane in the series. Figure 7 shows these membrane performance indicators for different ZSM-22/PES NF membrane with zeolite loadings of 0–0.75 wt.%. 

Therein, Figure 7a indicates that, as the loading of ZSM-22 was increased, the membranes showed increasing flux response at the same pressure. Thus, membranes with high hydrophilicity showed relative high flux in line with expectations. Moreover, the flux recovery ratios were assessed by using BSA as a model foulant (Figure 7b). The unmodified polyethersulfone membrane, which exhibited a greater contact angle due to its inherited low hydrophilicity, attained the lowest flux recovery ratio (FFR). This was a typical hydrophobic membrane and followed its reported character [78,79,80,81]. This behavior suggests that the membrane was fouled during BSA rejection and backwashing could not restore the membrane performance/flux. However, the FRR was observed to increase with increasing ZSM-22 wt.% loadings (as manifested in Figure 7b), suggesting the improved fouling resistance of the PES composite membrane upon zeolite addition. As shown in Figure 7a at higher wt.% loadings, the incorporation of porous zeolitic materials has led to low resistant water permeability in agreement with the obtained FFR in Figure 7b. This might be due to additional flow paths presented by the porous nanoadditives, thus increasing the tortuosity of the matrix [82,83]. 

In this study, the solutes used for membrane rejection were inorganic salts (NaCl and MgCl_2_), representing mono and divalent salts, respectively. The two salts were selected to better assess the influence of the porous and negatively charged ZSM-22 zeolite on the behaviour of NF/RO membranes for salts. This assessment led to cheaper water softening applications using lower applied pressure (<1 bar) in the NF or RO membrane systems. The interaction of these solutes with the composite membrane matrix containing negatively charge nanoadditives, as opposed to size exclusion, was also a possible mode for membrane rejection [84,85]. The relatively high charge on divalent Mg ion means it is strongly attracted to the negatively charged membrane surface than the monovalent Na ion, resulting in the different observed rejection profiles. As displayed in Figure 7c, the composite membranes attained about 50% of NaCl rejection, while the best rejection (≥65%) for MgCl_2_ was for membranes with a ZSM-22 loading of 0.3 wt.%. The rest of the composites rejected between 60% to 55% of the salts. This shows that the electrostatic repulsion of Mg^2+^ by the membrane was much higher than that of Na^+^, hence why the high rejection of MgCl_2_ was observed. It can be concluded that the composite membranes in this study possess negatively charge surfaces due to the AlO_4_ tetrahedra interactions with the SiOH of ZSM-22 nanoadditives [84,86,87].

## 4. Conclusions

Zeolite ZSM-22 material was successfully synthesised and fully characterised by XRD, BET, FTIR, SEM and TEM. The zeolitic materials were incorporated into PES membranes through phase inversion. Membranes were characterised by XRD, FTIR, WCA and SEM and their performance in relation to pure water flux, salt rejection and protein fouling, i.e. flux recovery ratio was assessed. The increased hydrophilicity with increasing amounts of ZSM-22 resulted in membranes having increased flux and flux recovery ratios or increasing protein (BSA) fouling resistance. The solute rejection for the monovalent NaCl was insensitive to the nanoadditive loadings, while the divalent salt, MgCl_2_, reached a maximum before decreasing with increasing loadings. An approximate minimum rejection of about 55% was achieved upon using these materials, though a further assessment of the performance of the membrane still needs to be carried out.

## Figures and Tables

**Figure 1 polymers-12-01446-f001:**
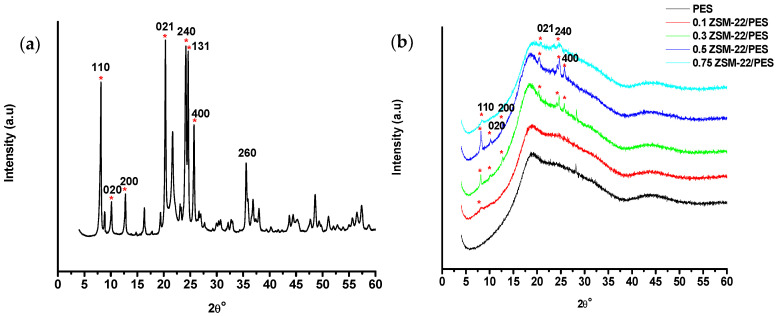
XRD patterns of (**a**) ZSM-22 material and (**b**) ZSM-22/polyethersulfone (PES) nanofiltration/reverse osmosis (NF/RO) membranes, respectively, prepared via hydrothermal synthesis approach and direct phase inversion method.

**Figure 2 polymers-12-01446-f002:**
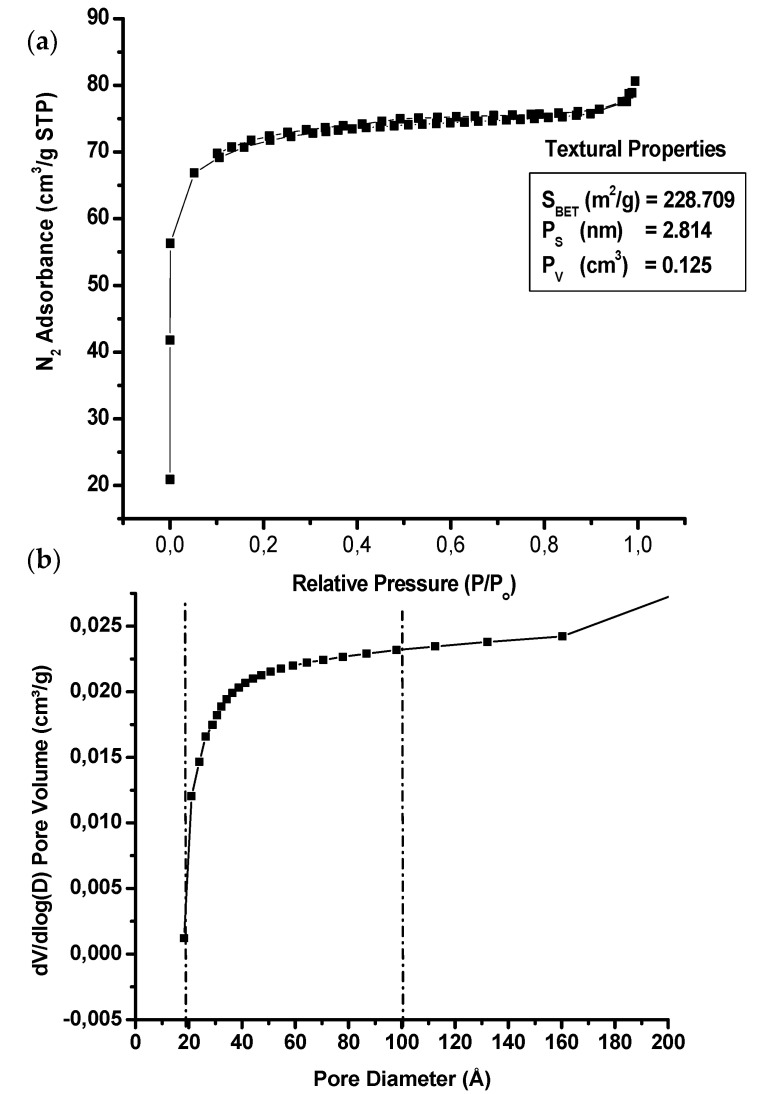
BET Isotherms (**a**) N_2_ adsorption–desorption uptake and (**b**) calculated BJH pore distributions of a hydrothermally synthesised ZSM-22 material.

**Figure 3 polymers-12-01446-f003:**
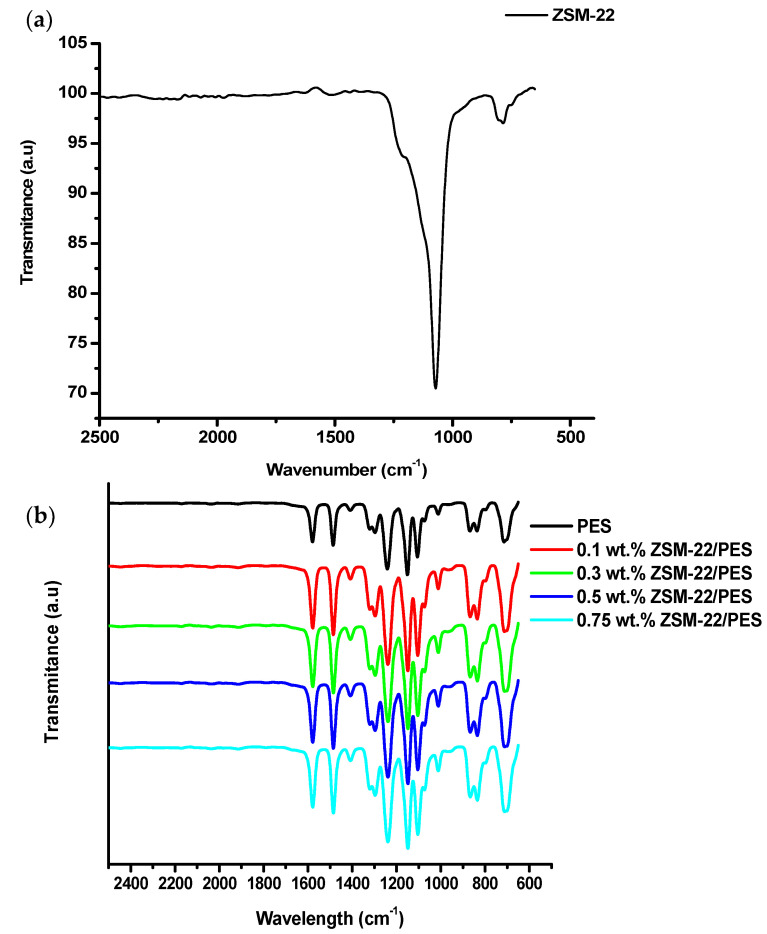
(**a**) FTIR Spectra of ZSM-22 material prepared via hydrothermal synthesis method, (**a**,**b**) ZSM-22/PES NF/RO membranes prepared via direct phase inversion method.

**Figure 4 polymers-12-01446-f004:**
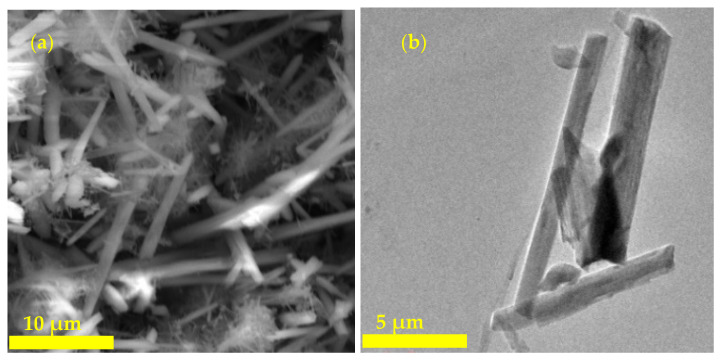
SEM (**a**) and TEM (**b**) micrographs of ZSM-22 zeolite material hydrothermally synthesised using tetraethyl orthosilicate (TEOS) source at Si/Al 60.

**Figure 5 polymers-12-01446-f005:**
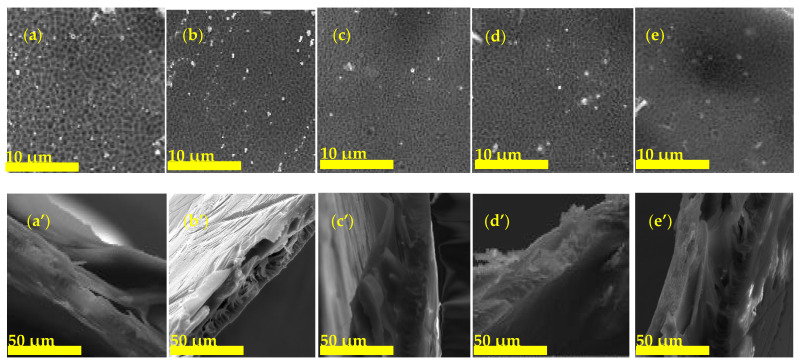
SEM surface and cross-sectional’ micrographs of (**a**) PES and (**b**–**e**) ZSM-22/PES membrane materials prepared using different ZSM-22 content: (**a**,**a’**) PES, (**b**,**b’**) 0.1 ZSM-22/PES, (**c**,**c’**) 0.3 ZSM-22/PES, (**d**,**d’**) 0.5 ZSM-22/PES and (**e**,**e’**) 0.75 ZSM-22/PES.

**Figure 6 polymers-12-01446-f006:**
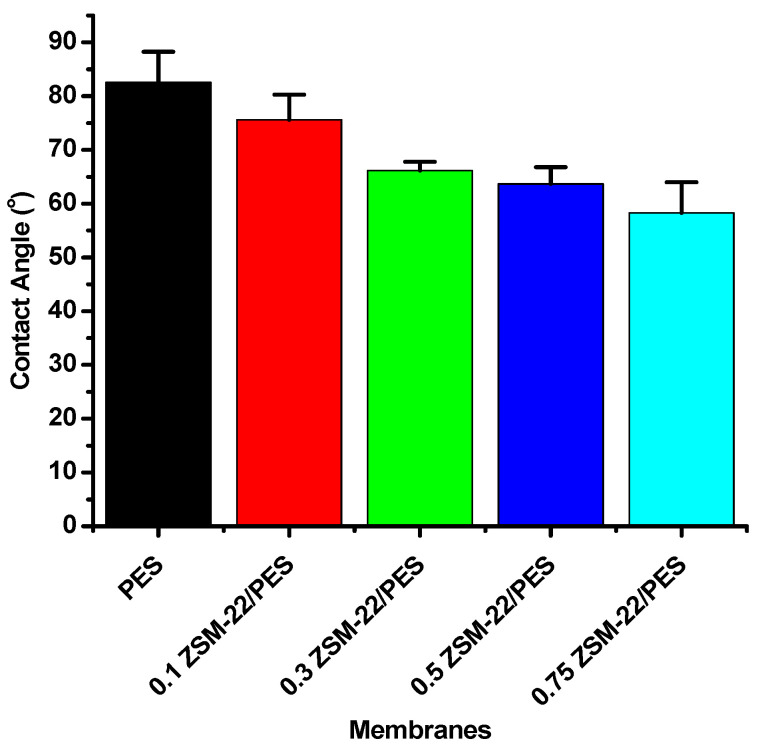
Contact angle measurements of ZSM-22/PES membrane materials prepared using different ZSM-22 zeolite wt.% loadings.

**Figure 7 polymers-12-01446-f007:**
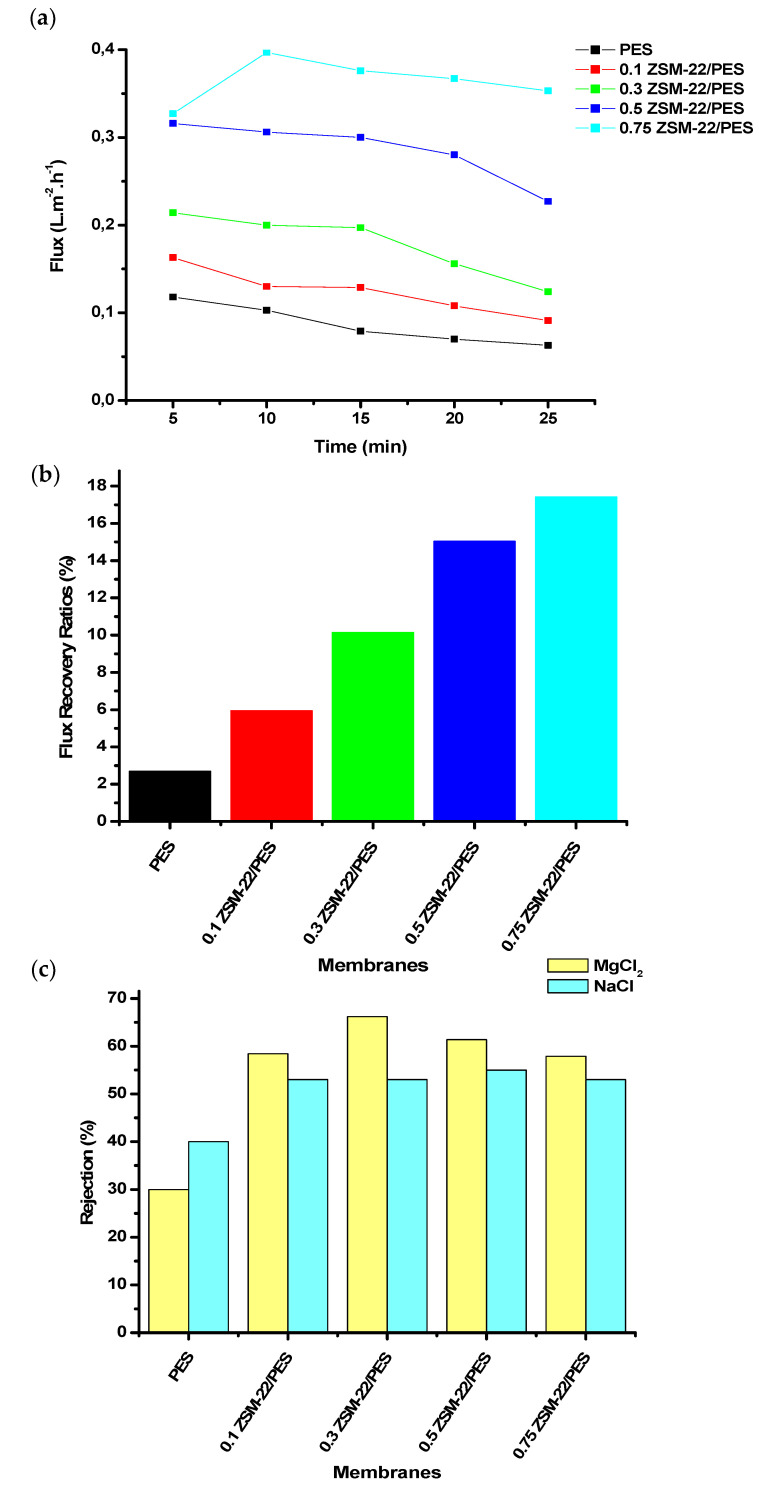
Performance evaluation. (**a**) Pure water flux, (**b**) flux recovery ratio and (**c**) salt rejection analysis graphs of differently loaded ZSM-22/PES membrane materials prepared using ZSM-22 as a nanoadditive.

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
