# Peer review of "Development of ZSM-22/Polyethersulfone Membrane for Effective Salt Rejection"

_polymers, 2020, doi:10.3390/polym12071446_

Round 1

Reviewer 1 Report

Manuscript Title: Development of ZSM-22/PES Membrane for Effective Salt Rejection

Journal Title: Polymers

Authors: Nyiko Chauke, Richard Moutloali *, James Ramontja

Coresponding author: Richard Moutloali

Manuscript ID: polymers-813260

 - The importance of the topic and whether it is suitable for publication in this journal – Yes, I feel the topic has high importance and suitable for publication in Polymers.

-The originality of the work – This work in novel and original

- Does the manuscript use sufficient references, appropriate language and styles? – After careful evaluation of the references I feel that it’s OK.

- Does the title of the manuscript is appropriate? – Yes, the title is appropriate.

- The abstract adequately summarizes the content of the manuscript? – Yes, Abstract is OK. Authors highlighted the main idea on the paper.

- The findings of this manuscript are correctly interpreted? Yes, correct.

- The quality of figures and illustrations is acceptable? Yes.

The manuscript is devoted to synthesis of Zeolite ZSM-22 material and incorporation them into PES membranes for water treatment. Zeolite and prepared membranes were characterized by advanced methods such as XRD, FTIR, BET, SEM, TEM.  The method allows increasing hydrophilicity of the membranes and increase performance as well as fouling resistance against BSA. The manuscript is well-organized, contain advanced methods of characterization. Topic of the manuscript is relevant and interesting. Based on these points, this manuscript can be recommended for publication in the journal after major revisions with due consideration of the specific comments below.

1) Page 3, line 95 “The pre-dissolved TEOS (29 mL) in deionised water (69 mL)”. However TEOS is not soluble in water. TEOS slowly react with water.  Please also describe this procedure in more details. What time is TEOS-water mixture was preconditioned before dropping? How long TEOS-water mixture mixture was adding to the solution?

2) Page 3, line 99. Why crystallisation was performed at 160oC for 144h? 

3) Page 3, line 101. Please use oC or K in the manuscript. 

4) Page 3, line 125. Please indicate power of sonication.

5) Page 4, line 140. Please indicate temperature in refrigerator where solutions were kept before use. Why solutions were kept in refrigerator? Are there any recommendations (references) for this?

6)  Figure 1. Please add crystal indexes.  (a) and (b) are cutted off. Please correct.

7) When describing XRD, references confirming observations should be added where appropriate.

8) Figure 2b should be reconstructed. The described phenomena are very poorly visible. It is suggested to increase the size of the picture and reduce the thickness of the lines Also please indicate main bands in the figure.  This will help the reader to verify the phenomena you describe.

9) References which can confirm discussion of FTIR spectra should be added. 

10) Figure 6. Please put error bars. It is also suggested to add photo of the drops.

11)  Figure 7. Please increase the picture. Please increase the font size. 

12) Please make a more detailed analysis of the obtained results on performance and salt rejection of prepared membranes with previously described in the literature.

13) It is suggested to analyze distribution of zeolitic materials in PES membranes, for example by EDX analysis.

14) It is suggested to added references from Polymers related to the topic of the manuscript.

15) It is suggested to improve the Introduction part by new relevant literature and I suggest to consider the following references: 10.1016/j.seppur.2019.115694; 10.1016/j.pnucene.2019.103128; 10.1016/j.matchemphys.2017.11.006. 

Reviewer 2 Report

In this manuscript, the authors investigated the preparation of zeolite incorporated PES membrane for salt rejection. The results are clearly presented. However, the research area is not new, and actually, the field of nanocomposite membrane for water purification is already a crowded one. Most importantly, there is a major issue in this study. Normally, the nano-/micro-additives used in preparing pressure-driven membranes using the non-solvent induced phase separation method are “pore-opener”. The addition of the zeolite may increase the pore size of the membrane, and that is NOT good for the performance. There is no such discussion in this study. Therefore, I do not recommend publishing this manuscript on Polymers.

Reviewer 3 Report

This manuscript provides some information about the effects of the addition of ZSM-22 on the PES membrane performance. In this work, the authors fabricated  ZSM-22 zeolite material and ZSM-22/PES membranes, evaluated characterisation and membrane performance of the resulting materials with XRD, BET, FTIR, TEM, SEM, contact angle and dead-end cell. Moreover, they presented a quite thorough analysis of the morphology, hydrophobicity and hydrophilicity of the membranes,flux and rejection analysis. The paper is well organized and findings are consistent. I suggest its publication in Polymers after addressing the minor issues indicated below. Some particular points to be taken into account are listed:

1 Line 90-92: When KOH and Al2(SO4)3·18H2O was dissolved in deionised water, have the authors followed a sequence? And does solution A precipitate during its dissolving process? If yes, how to solve it?

2 Line 103-104: Why is NMP chosen  as the solvent instead of DMAc and DMF, which are less toxic and inexpensive?

3 Line 111-112: When the membrane was kept in a DI water bath at room temperature for 24 h, is the DI water replaced during Membrane soaking process?

4 Line 221-222: Can the authors explain the effect of HDMA used as a structure directing agent on the membrane?

5 Line 287-289: What effect did the porous and negatively charged ZMS-22 zeolite on the behavior of NF/RO membranes for salts? Please explain briefly.

Round 2

Reviewer 1 Report

Manuscript could be recommended for publication in this form

Author Response

Not applicable

Reviewer 2 Report

Thanks very much for the authors' response. I appreciate the revisions made by the authors. However, I'm afraid that the issue I initially mentioned in the first round review still exists.

First, the structure of the membrane significantly depends on the preparation conditions. It is not convincing to just use results from other studies in the literature to support the claim in this study. In ref [70], figure 3, the PES membrane without zeolite is dense while adding zeolite results in asymmetric membrane with finger-type pores. However, in the other two ref [68] and [69], the pore structure is different from that in [70] even though all of them added zeolites. So, in this study, authors are recommended to add more characterizations (SEM, for instance. The current SEM pictures in Figure 5 do not show clear information) and discussion to support the corresponding conclusions.

The authors added “As such, incorporation of ZSM-22 into the PES polymer matrix resulted in reduced membrane pore sizes while increasing pore density compared to pristine PES membrane surface”. However, this is merely speculation without convincing characterization support.

Round 3

Reviewer 2 Report

Thanks very much for the revision and response. I think it's ready to be published on Polymers now.